# Mothers' Perception about Mediated Learning Strategies Used in the Home Environment for Supporting the Transfer Ability in Children with Down Syndrome: An Exploratory Investigation

**Francesca Granone** [1,*], **Martin Stokke** [2], **Sandra Damnotti** [3], **Chiara Chicco** [4] **and Enrico Pollarolo** [5]

1   Department of Early Childhood Education, University of Stavanger, 4021 Stavanger, Norway
2   Department of Teacher Education, Art and Culture, Nord University, 8049 Levanger, Norway; martin.stokke@nord.no
3   International Center for Studies on Educational Methodologies (ICSEM), 10121 Torino, Italy; sandradamnotti@gmail.com
4   Mediation Azione per la Ricerca sulla Ristrutturazione Cognitiva e l'Apprendimento Mediato (ARRCA), Via dei Mille, 48, 10123 Torino, Italy; cchicco@hotmail.com
5   Norwegian Centre for Learning Environment and Behavioral Research in Education, University of Stavanger, 4021 Stavanger, Norway; enrico.pollarolo@uis.no
*   Correspondence: francesca.granone@uis.no

**Abstract:** Down syndrome (DS) is the most identified genetic form of disability. Individuals with DS have cognitive and linguistic impairments that vary from severe to mild, although they may show strengths in imitation, social learning, and the use of body language. Many studies have shown that early interventions for sustaining the development of children with DS (physically, linguistically, and cognitively) provide important results, enhance their abilities, and improve their life. Children with DS benefit when supported in their linguistic and cognitive development in the family context, for example, through the ability to compare and relate objects or situations (i.e., transfer competence). Although many studies have shown mediated learning strategies applied in a home environment to enhance children's analogical, analytical, or inferential thinking, including for people with disabilities, there remains a lack of studies of these strategies for children younger than three years of age. This is in contrast with studies that present analogical thinking (i.e., transfer ability) as one of the fundamental approaches developed before the age of three. The present study aims to highlight mothers' perceptions about mediated learning strategies used in the home environment to support the transfer ability in children with DS. Five mothers of children with DS who were younger than 3 years of age were involved in the project. They carried out two activities with their child, participated in an online workshop, and answered a questionnaire. The questionnaire's answers were subjected to thematic analysis. This analysis revealed four main themes: motivation for learning and applying mediational strategies in a family environment, mediational styles applied during activities, language and cognition, and the family as a part of the educational environment. Based on the findings, new lines of future research are suggested.

**Keywords:** mothers; children with Down Syndrome; Feuerstein; mediation; transfer ability; cognitive functions; linguistic and cognitive skills

## 1. Introduction

Down Syndrome (DS) is the most commonly identified genetic form of disability [1]. Children with DS can show impairment in expressive grammar and vocabulary, although receptive vocabulary is often less impaired [2,3]. Particular aspects of declarative or recall memory are usually also impaired [4]. Further, children with DS have a deficit in the verbal loop of short-term memory [5], although they have a good ability to understand through

the visual part of short-term memory [6]. Imitation, social learning [7], and the use of gestures to communicate [8] can be considered as strengths.

DS can affect communicational, intellectual, or motor skills. Various programmes and different approaches have been introduced to sustain and enhance each person's best abilities [1,9]. In case of disability, a timely intervention in supporting all of the child's functions is considered important, regardless of whether the disability is related to communicational, intellectual, or motor skills [10–12]. Supporting children with DS in the development of these skills can be crucial for promoting self-esteem and, consequently, inclusion [13]. Studies have analysed how people with DS can be supported in their language development [14] or cognitive development [15] to, for example, make comparisons and relate objects or situations [16,17]. This ability has been called transfer [18], bridging [19], or analogical thinking [20]. In this article, we refer to it as "transfer" because it is related more generally to critical thinking [18], whereas bridging is specifically used in the Feuerstein approach [19] and analogical thinking is usually considered to be more connected to a mathematical approach [21]. Preschool teachers have identified it as the ability for children to transfer what they have learned to new experiences and settings. For example, children acquire building techniques in a sandbox, and then they can use these competences to construct new things with Lego [22].

The ability to transfer has already been identified as a key competence connected to critical thinking at preschool age [22]. Further, this has been discussed for both typical developing children [23] and children with disabilities [24,25].

Working with an adult who provides guidance and leadership throughout the learning experience can support analogical thinking [26] and have cognitive benefits [27,28], and active parental interaction is crucial for enhancing children's cognitive development [29–33]. Studies have mainly investigated the role of the mother as a mediator in a home environment [34,35], and many studies have shown the efficacy of the maternal role [36,37]. Although many studies have applied mediated learning strategies in a home environment to enhance children's analogical, analytical, or inferential thinking, including for those with disabilities [35], there remains a lack of studies of such strategies for children younger than three years of age. This contrasts with studies that have noted analogical thinking (i.e., transfer ability) as one of the fundamental approaches developed before the age of three [23]. In this light, the present study aims to highlight mothers' perceptions about mediated learning strategies used in the home environment to support the transfer ability in children with DS. This study investigates the perception of mothers and not the strategies used by them in practice because of the small number of participants involved (in fact, it is an exploratory investigation) and the fact that the way in which mothers perceive the mediation influences how they apply it. It was therefore considered necessary to first investigate mothers' perception and only then investigate the possible approaches to mediation. This article does not report on the methods put into practice by mothers.

## 2. Theoretical Framework

### 2.1. Feuerstein's Theory

The present study is grounded on Feuerstein's theory [15,38] and, in particular, on the approach called Mediated Learning Experience (MLE). In fact, among other cognitive developmental models such as those reported in pioneering works with cohorts of parents of young children with DS [33] or clinical works on parents' responsive teaching in early intervention [36], Feuerstein's approach can provide both a set of principles and a guiding theory [34] that can be applied at a micro level to observe specific parent–child interactions [39]. Feuerstein's approach is centred on an optimistic vision concerning the possibility of modifying a human being stably through the education of the cognitive processes that allow the modification of the cognitive structure. The basic theoretical matrix is that of constructivism linked to Piaget and Vygotsky [40,41], where knowledge is considered the effect of relationships between the individual and reality. Cognitive education is defined as a complementary education aimed at everyone, regardless of functional level,

with the aim of potentially improving cognitive aspects. In this approach, the mediator (e.g., teacher, parent, sibling) plays a fundamental role. Such mediators need training that allows them to understand their role [19].

In the MLE approach, the family is presented as a mediational unit, where problem solving and a support structure for the child's effort exist [42]. The application of MLE to children with learning disabilities or learning difficulties to enhance their thinking ability has been extensively studied, and the benefits of this approach have been presented clearly [15,34,38,43]. MLE has been applied and specifically developed in different ways, depending on the age of the children receiving the support [19]. The Feuerstein Instrumental Enrichment Standard Program is designed for children from eight years of age [43], whereas the Feuerstein Instrumental Enrichment Base Program is designed for children from two years of age [44]. These approaches are based on activities that are presented to children to stimulate their emerging cognitive functions (CFs) and thereby enhance their development [45]. Haywood's Bright Start approach is based on the same method but is more related to group activities and play for children from three years of age [46]. The reliability of these methods in enhancing children's cognitive development is well known [43,46,47]. A mediational approach related to play or everyday situations has been considered useful [48]; however, specific activities have not been described in detail for children younger than three years of age. Play and problem solving in an everyday situation are two important aspects of Norwegian preschool education [48,49], and the present project's aim is to investigate this research area.

### 2.2. Cognitive Functions

An important aspect of the Feuerstein approach is the possibility to identify the CFs defined as emerging in each child and to specifically work on them by relying on those functions that are already consolidated [19]. Feuerstein considered that CFs contribute to mental acts (and, consequently, real acts). Therefore, the deficiencies in the acts can be traced back to difficulties in the requisites of the acts themselves and therefore also in the CFs. This implies that CFs can be observed by observing the acts and that by working on the CFs through mediation, the acts can be improved [19].

In Table 1, CFs are identified through abbreviations, adapted from the classification presented in the Pas Basic description in relation to the three phases of the Mental Act [50–52].

**Table 1.** Cognitive functions.

| Input | Elaboration | Output |
|---|---|---|
| CF1: Systemic exploration of a learning situation. | CF6: Ability to experience and define a problem. | CF13: Controlled and planned expression of thought and actions. |
| CF2: Precise and accurate understanding of words and meaning. | CF7: Ability to think about information stored in one's brain. | CF14: Precise and accurate use of words and concepts. |
| CF3: Well-developed understanding of space concept. | CF8: Ability to automatically make comparisons. | CF15: Mature communication. |
| CF4: Well-developed understanding of time concept. | CF9: Ability to select relevant cues. | |
| CF5: Ability to use information from more than one source. | CF10: Ability to relate objects and events to previous and anticipated situations. | |
| | CF11: ability to automatically summarise information. | |
| | CF12: ability to engage in planning behaviour. | |

For the transfer ability, two CFs are particularly relevant: CF8 (ability to automatically make comparisons) and CF10 (ability to relate objects and events to previous and anticipated situations). The present article focuses on these CFs. Specifically, parents were asked to reflect about CF8 during activity one and CF10 during activity two rather than about

all the cognitive functions that the activities aroused (Appendix A presents the Cognitive Maps associated with the two activities).

### 2.3. Mediational Criteria

The MLE approach outlines different intervention criteria that allow children and young people to be supported in their development regardless of their functional level [53]. The interventions are based on activities that can be described through a Cognitive Map. The Cognitive Map is a conceptual framework that helps analyse the interaction between a learner's cognitive acts and the elements of the activity that are relevant to these acts [51]. This process of analysing a task to identify the cognitive operations is called "task analysis" [51,54]. A mediator must be familiar with each of the cognitive operations required to carry out an activity to mediate the learning process and encourage the learner to think analogically (i.e., to transfer what has been learned to a new task that requires similar cognitive operations to the previous one).

As shown in Table 2 (data from [55]), the Feuerstein approach can be described through 12 criteria, of which three are considered universal (intentionality and reciprocity, transcendence, and significance) and nine are additional [56].

**Table 2.** Mediational criteria.

| Mediational Criteria | |
|---|---|
| Intentionality and reciprocity | The mediator shows his intentions to the child, with the aim of involving him/her in a mutual process of learning. It is fundamental that the mediator verifies the child's understanding about the goal of the activity. |
| Transcendence | The mediator helps the child to use the acquired knowledge in new situations. |
| Significance | The mediator explains the intention of the activities so that they make sense. He/she should raise child's interest in the task itself. |
| Planning and achieving objectives | The mediator helps children to set their objectives and to approach them with perseverance, patience, and hard work. |
| Sense of competence | The mediator has the responsibility of creating an environment in which children can develop high self-esteem and self-confidence, as a means for acquiring the necessary abilities and strategies. This can be realised by adapting activities according to children's interests and ages. |
| Awareness of change | The mediator helps children in becoming aware about the fact that they can change, improve, and strengthen their cognitive functions and their behaviours and strategies in order to achieve their objectives. |
| Novelty and complexity | The mediator presents challenges to the children. The activities organised should be sufficiently difficult to provide a challenge, but still achievable. |
| Active participation and shared conduct | The mediator helps children in developing logical and systematic processes to solve problems by using previously acquired knowledge, expressing it through reasoning procedures. |
| Individuality and psychological difference | The mediator strengthens the importance and the richness of being unique person with different thoughts and different learning processes. |
| Sense of belonging | The mediator promotes teamwork and reinforces reciprocal support, tolerance, respect, confidence, and empathy. |
| Optimistic awareness | The mediator underlines that is not important how difficult an activity seems to the children; they have to be aware that all of them will be capable of doing it. |

## 3. Materials and Methods

### 3.1. Participants

To investigate mothers' perceptions about the mediated learning strategies used in the home environment to support the transfer ability, five mothers of children with DS were involved in the project.

The five children with DS were below three years of age; they were observed playing in their home environment by their mother during the study. Table 3 reports the participants' characteristics.

**Table 3.** Participants.

| Mother | Mother's Age | Child | Child's Age |
|--------|--------------|-------|-------------|
| Amalie | 43 years old | Tor | 2 years old |
| Rita | 45 years old | Bente | 2.3 years old |
| Nora | 29 years old | Oliver | 2.5 years old |
| Ingunn | 32 years old | Marie | 2.5 years old |
| Anita | 35 years old | Elise | 2.9 years old |

All names listed above are pseudonyms. The mothers were involved in various activities: they carried out two activities with their child ("matching the figures" and "matching and pairing socks"), participated in an online workshop, and answered a questionnaire. Participation was voluntary; however, potential participants had to satisfy some specific characteristics to become involved in the project:

- The mothers who participated in the research project had to be trained in the Feuerstein approach and had to know the six signs describing the animals in the pictures.
- The children had to be included in the ECEC institutions program, which included a specific type of sign language (a basic version called "from the sign to the word" (this is called "Tegn til tale" in Norwegian and is a specific approach used with children who have impaired speech to help them to communicate during the development of speech) where just some words in a sentence are presented through signs) and had to be able to express the signs for the animals in the pictures.

### 3.2. Workshop and Questionnaire

3.2.1. Matching Figures

The first suggested activity suitable for the specific level of the children's development was "matching the figures". This activity featured a table with six images and the corresponding six images available for the child individually cut out (Figure 1). The child was invited to label the figure (by recognising it and assigning a name verbally or through signs) and to match the figures. The recognition of analogies is important because it is the basis of the subsequent development of analogical thinking [57].

In ECEC institutions, the activity was presented to develop the lexicon [14].

In contrast, at home, the activity was used to observe the acts (and, consequently, the cognitive functions that could be associated with the acts). As described in the corresponding Cognitive Map (Table A1), different cognitive functions can be observed. The mothers received the indication of observing in particular acts that could be related to "CF8: ability to automatically make comparisons". This task was simple to understand and to perform because the mothers were trained in the Feuerstein approach.

3.2.2. Matching and Pairing Socks

In accordance with the importance of helping children reinforce their competences through everyday activities [47]—for example, through activities related to practical life [58]—the second chosen activity was "matching and pairing socks". Each pair was labelled with the same animals presented in the activity "Matching the figures" (Figure 2). All pairs had the same colour to guide the child in matching and pairing with just the picture. This activity presents many useful aspects for the purpose of this exploratory project:

- It engages children in an activity that falls within the Montessori activity "tidying up." It empowers the child and stimulates his/her fine motor skills [59].
- It allows children to use the knowledge developed in the first activity, which is similar but not identical, to facilitate the transition from knowledge to competence [59].

- It allows working specifically on the cognitive function "CF10: ability to relate objects and events to previous and anticipated situations", allowing the child to reinforce the transfer competence [57]. In fact, if additional cognitive functions have also been observed through this activity, as described in the correspondent Cognitive Map (Table A2), the mothers had to observe acts related to transfer competence. This includes relation to CF10 as well as the use of this competence to solve the new pairing activity.

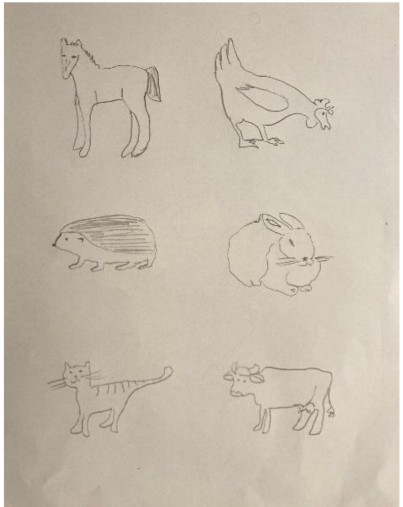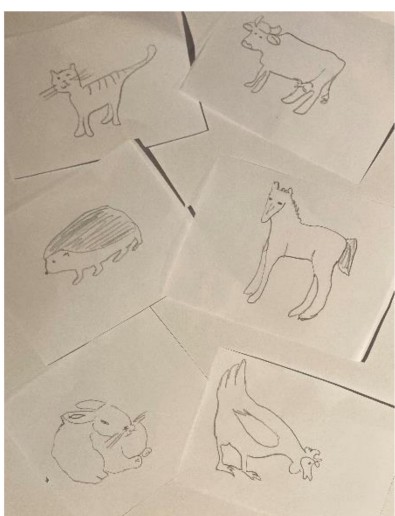

**Figure 1.** Activity of "matching the figures." (Adapted from the activities included in the Karlstad-model program, edited by Johansson [14]).

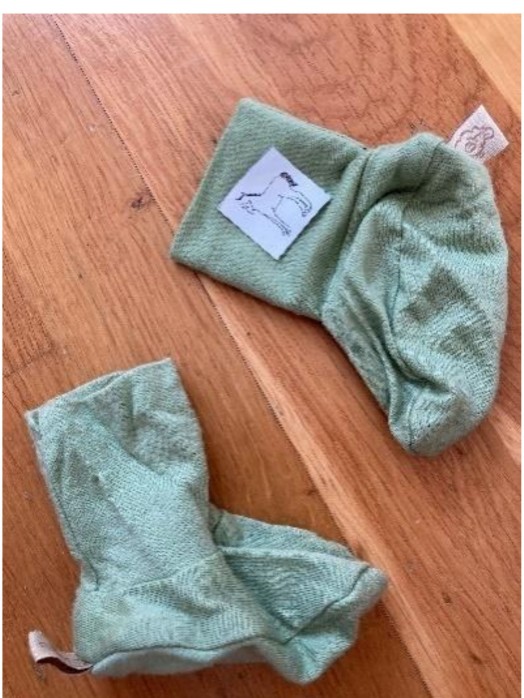

**Figure 2.** Activity of "matching and pairing socks".

### 3.2.3. Workshop and Questionnaire

The mothers were invited to a two-hour workshop online, where they received the Cognitive Map (Appendix A) associated with each activity, to reflect on their experience and observations. Then, they received a questionnaire (Appendix B) that they had to compile and send to the researchers before the last part of the workshop. During the last

part, the mothers were invited to a discussion about the activities, their experience, the cognitive maps, and the whole project. The last question of the questionnaire related to suggestions must be discussed together.

### 3.3. Procedures

Each mother received information from the ECEC teachers about the communication development of the child at the beginning of the project, with particular focus on the knowledge of the specific animals' names represented in the activities. In the ECEC institution, "matching the figures" was presented to the child three times a week (Monday, Wednesday, and Friday), with a focus on language development. The time devoted to matching the figures was 15 min per session.

The mothers used the same activity at home two days a week (Tuesday and Thursday) during the same month, with a focus on the cognitive function CF8 (taking notes concerning the child's behaviour). The time devoted to this activity in the home environment was also 15 min per session.

Mothers had to choose the mediational criteria that they considered most suitable to the situation. Mothers received no instructions about the acts that could be observable in connection with CF8 and CF10. Both the mediational criteria and the acts observed were part of the questionnaire.

After this month, the mothers focused on the second activity in the home environment for 15 min per session, twice a week for two weeks, with observation of observing acts related to CF10.

The mothers were invited in taking notes about their observations. These notes were shared with the researchers before the workshop.

Then, researchers and mothers participated in a 2 h online workshop to discuss observations, Cognitive Maps (Appendix B), and answers to the questionnaire (Appendix A).

### 3.4. Analysis

The questionnaire's answers were subjected to thematic analysis [60]. The first step of the analysis involved familiarisation with the data and note-taking during multiple readings. Then, the data were systematically analysed, beginning with data coding. An inductive approach was used, with the awareness that it is impossible to be purely inductive because researchers always bring their own notions to data analyses [60].

The data were coded according to four elemental coding methods [61]: descriptive, in vivo, process, and concept coding. These elemental methods were effective in assigning labels to the data. Then, consultations between the researchers were conducted to identify the final themes.

The parents spoke in Norwegian. Quotations from the transcripts are selected as illustrative examples for themes and patterns. The quotations have been translated from Norwegian to English with considerable effort to preserve the participants' original meanings. At times, however, it was necessary to modify the sentence structure for better readability.

### 3.5. Ethics

Ethical guidelines have been respected. The invitation to participate in the study was sent through the network of a volunteer organisation. All parents who sent a message of interest received information about the study. Informed consent to participate in the project was obtained from the parents. In particular, they were assured that the material would be anonymised in all publications relating to the project and that the data would be treated with a high level of confidentiality. Five parents responded positively. The ethical guidelines of the Norwegian Social Science Data Services, which gave permission for the project, have been satisfied.

## 4. Results and Discussion

The data analysis resulted in four main themes:

(1) Motivation for learning and applying mediational strategies in a family environment.
(2) Mediational styles applied during activities.
(3) Language and cognition.
(4) The family as a part of the educational environment.

### 4.1. Motivation for Learning and Applying Mediational Strategies in a Family Environment

Participants had diverse motivations for learning and applying mediational strategies, as illustrated by the following quotes.

Four parents out of five answered that they had personal reasons for involving themselves in a learning process. Amalie said, "*I have studied this method because I had understood that it was a way for better communicating with my son. This is what was written in the literature. I think that I have understood more, honestly. I have more . . . hope . . . or maybe trust . . . yes trust. I behave with him in the same way as I do with his brother now.*" Ingunn said, "*I studied the theory in general, and after, I attended a course for FIE basic. I did it because I thought that if I can have more knowledge, I can help him better. And I feel better. And I see him better.*"

Nora studied the MLE for work reasons, but she evaluated the application of the methods in the family environment.

These considerations can be analysed in light of Feuerstein's approach. This approach is centred on an optimistic vision concerning the possibility of modifying a human being stably through the education of the cognitive processes that allow modification of the cognitive structure. Cognitive education is defined as a complementary education aimed at everyone, regardless of functional level, with the aim of potentially improving cognitive aspects [19].

Additionally, other viewpoints can be introduced to enrich the discussion. Amalie's consideration is in accordance with research that shows that families appear to adjust well to the challenges of rearing a child with DS [62] and that parents of children with DS may feel equally rewarded compared to parents of same-aged typical children [63]. Additionally, this connects to the studies of Rondal [64] that demonstrate how the mothers of children with DS can adapt their language to the linguistical level of their children in a manner similar to that of the parents of typically developing children. In accordance with what just presented, Ingunn stated, "*Yes maybe . . . but I could understand her, communicate with her. So maybe I have learned something . . . that I can find a dialogue with her, that I can observe her . . . thinking. I understood how important it is to support her, starting from what she knows.*"

Ingunn's statement helps reflect "parental efficacy". Parental efficacy can be described as parents' belief in their ability to perform the parenting role successfully [65]. Parenting efficacy is considered an important factor for a positive perception of the child [66,67] and for lower levels of stress in families that include a child with an intellectual disability [68]. For parents of children with DS, self-efficacy reflects parents' confidence in having the skills to provide the necessary guidance to their child [69] and function as educators [70] to help them develop important life skills, such as linguistic and cognitive skills [71].

The mothers noted the importance of applying a mediational approach not just in specific activities, but in everyday situations (e.g., tidying up clothes). Rita said, "*I really think that was interesting to use it in a "real life work". Because as parents, we do it every day, but now I have seen how much thinking can be found behind each action.*" Similarly, Amalie said, "*I also used the transcendence, because I think that it is really important to help my son to connect to reality, not just to solve a problem here and now. How is it called . . . competence?*" The importance of connecting to reality and not just to structured activities is in accordance with the fact that both structured activities and free-play situations can be used as MLEs and that interventions through free-play and daily activities help develop the child's potential [43,72,73]. However, many studies have explored structured situations, and therefore, free-play situations deserve further research attention [35]. Anita noted an important aspect related to the second activity: "*She shows interest, concentration, she shows*

*happiness. Usually when I tidy up, I have not time for staying with her, but now we do it together. And she is improving her movements."* This can be connected to the specific choice made about the second activity. The second activity proposed in fact was chosen from among Montessori activities related to "tidying up" This group of activities is part of what are called "practical life" [59]. All the routine activities that adults carry out in practical life habitually can be interesting, and many children show a desire to imitate. Allowing the child to take part in such exercises is important because it can help him/her develop and improve both fine and gross motor skills.

*4.2. Mediational Styles Applied during Activities*

The mothers identified all the 12 strategies as important. Amalie highlighted that it is not precise to identify the most important one but instead to reflect on the fact that each person can have different mediational styles: *"Yes, I think that it is not correct to talk about more important strategies . . . but . . . more applicable. Intentionality, transcendence . . . but it is difficult to explain why . . . and now I wonder . . . is the same from every one of us? Me? The brothers? My husband?"* This is in accordance with literature that shows how the mediational approaches of mothers and siblings can be different in the family environment [24,25]. In relation to the mediation style applied by the fathers, few studies have been reported [35].

Among all mediational strategies, the mothers participating in the exploratory study identified some strategies that seem more applicable for them during the proposed activities. Nora identified the mediation of meaning and transcendence as the most relevant one: *"Usually mediation of meaning. Specifically, here I can say mediation of meaning but also transcendence."* Anita pointed to bridging as an important strategy: *"I used bridging. Helping her to think about reality, to connect with reality."* Literature shows that, in particular, three mediation strategies have been highlighted as significant for enhancing the child's cognitive modifiability, namely the mediation of meaning, mediation of transcendence, and regulation of behaviour [72]. Studies of typically developing children highlighted the regulation of behaviour as the main mediation strategy applied by siblings [73]. Amalie confirmed the importance of transcendence, but also pointed to the strategy called "Intentionality and reciprocity": *"I have thought a lot about this question. I can say . . . I was really focused about "Intentionality and reciprocity". I wanted to have his attention, to start a communication with him. He can communicate a lot but I miss the sound . . . so I have to say that I am learning how to support him, but not too much . . . . I used also the transcendence [ . . . ]."* Intentionality and reciprocity have been presented in literature as a dominant strategy when the disability of the child impairs communication. However, it can result in a negative prediction for the child's psychological resilience [72] because the mother provides a higher level of focusing to overcome the child's difficulties.

*4.3. Language and Cognition*

Two questions were specifically related to language and cognition (i.e., transfer ability). Rita's answer shows that she is considering the importance of supporting children in learning how to think and how to express their thoughts through actions: *"I think a lot about language, the meaning of the words. Transfer means that you take something from somewhere and use it in another situation. So, I help my daughter to learn something, and after that I help her to use it in another situation. I really think that it was interesting to use it in a "real life work". Because as parent we do it every day, but now I have seen how much thinking can be found behind each action. [ . . . ]. She (Bente) shows that she has understood the task more through signs than through overlapping the figures. Now I think: what is most important? To communicate or to think? Should I believe more in her?"* Nora also noted the importance of labelling: *"He recognises all 6 (I received a message from the ECEC institution that he is improving with labelling) and this... this helps . . . ".* Similarly, Amalie said, *"It is clear that he wants to compare the figures. But he shows that the comparison is not about figures, but about the content of the pictures, because he "names" them through signs."* These answers can be seen in accordance with Chomsky's viewpoint about language and cognition [74]. According to this vision, children acquire

their first words by placing meaning upon familiar objects. For example, the word "cat" has no meaning to a child until he or she associates "cat" and the object that is a cat. When he/she understands this, he/she will be able to recognise and label it and will understand how to use it to communicate (verbally or non-verbally through sign) [75].

Anita highlighted another important aspect: "*I have understood that transfer is the ability to use what was learned in a previous context to solve new problems. So this is not just C10, is C10 used in act. A sort of . . . how is it called? Analogical thinking. You use the analogy. You define a structure in the reality, find a solution, apply it to a new situation. Those approaches seems really similar to me.*" Similarly, Ingunn said, "*These activities helped me as a mother to see my daughter think, to let her express hers though through actions. To see her logic.*" Analogy has been defined in literature as the core of cognition, and the ability to make analogies has been identified as the root of all our concepts because it makes sense of the new and unknown in terms of the old and known [76].

### 4.4. Family as a Part of Educational Environment

All mothers expressed satisfaction in being involved in the whole process of the research project and not just as an object of the research. Amalie said: "*I really think that we received recognition for our role as educators. Everyone talks about the educational role of the family but being invited to be part of the discussion in a research project is . . . is a step forward. It is about recognizing that we bring a different knowledge that other educational figures don't have. Confronting and discussing different points of view really leads to the good of the child.*" To better understand this answer, it is important to consider the Norwegian environment, where terms like "anerkjennelse" ("acknowledgment") and "medvirkning" ("cooperation") are keywords in the society, especially in the educational field [77]. The family is considered part of the educational team that sustains the child's development, and this is in accordance with a specific approach used in Scandinavia to support the language [78] as well as with Feuerstein's approach [34]. Nora said, "*We are an educating community, it was right to get involved in this way. It would have been even better to involve the kindergarten teachers as well.*"; this matches Hans Skjervheim's starting point. His theory, called the scientific theoretical self-understanding of the social sciences [79], points to an important position in society called the participatory position. A participatory position is developed when one subject is put together with the other, thus drawing attention to the phenomenon and engaging in his problem. In this situation, a symmetrical interpersonal relationship between the subject and the other is developed, and it is a key aspect in relation to the perspective behind special education [80]. As a consequence, both Anita and Ingunn affirmed their understanding of family's role as an important factor, pointing again to the importance of feeling competent about own parental efficacy [66,67]. They said, "*I realised how important the role of the family can be*" and "*I understood how important it is to support her, starting from what she knows.*"

## 5. Conclusions

In the present exploratory investigation, three aspects can be highlighted. First, MLE seems to be perceived by mothers as important in the family environment and also in a Norwegian context. The mothers revealed a propensity to preferentially use some of the 12 possible strategies, including choosing those that even the literature confirms as being most used in the family: the mediation of transcendence, mediation of meaning, and regulation of behaviour. These were considered by the mothers involved in the project as the most suitable to support their children in the application of transfer strategies.

Second, the mothers involved in the project considered it interesting to focus and reflect on the strategies that allow the development of children's transfer ability.

Third, the mothers expressed satisfaction in having been recognised as a fundamental part of the educational context. In particular, they highlighted the importance of being actively involved in the confrontation and discussion with other actors in the education of children so as to be able to work in synergy from different viewpoints.

The results of the present exploratory study cannot be generalised. Nonetheless, they can help to reflect on three possible elements that future research should investigate further. First is the role of the father as a mediator. While it is true that the mother has an important and evidently effective role in supporting the child's development, it could be interesting to investigate how fathers could enrich children's development through a mediational approach. Second, some daily activities could be more suitable for supporting children's transfer ability. The third aspect relates to the age of the children involved. Literature clearly shows that parents of young children with Down Syndrome are eager to support their children's linguistic and cognitive development. It could be interesting and important to investigate how to support families of older children or even adolescents with DS; for example, they could be involved in a research project that has the goal of supporting the children's logical ability.

**Author Contributions:** Conceptualisation, F.G., S.D., C.C., E.P.; methodology, F.G., S.D., C.C., software, F.G.; validation, S.D., C.C., M.S., E.P.; formal analysis, F.G.; investigation, F.G.; resources, F.G., S.D., C.C.; data curation, F.G., M.S.; writing—original draft preparation, F.G., M.S., E.P.; writing—review and editing, F.G., S.D., C.C., M.S., E.P.; visualization, F.G., S.D., C.C., M.S., E.P.; supervision, S.D., C.C.; project administration, F.G., E.P. All authors have read and agreed to the published version of the manuscript.

**Funding:** This research received no external funding.

**Institutional Review Board Statement:** The research approval was obtained from the University of Stavanger. The invitation to participate in the study was sent through the network of a volunteer organisation. All parents who send a message expressing interest were given information about the study. Informed consent to participate in the project was obtained from the parents. In particular, they were assured that the material would be anonymised in all publications relating to the project and that the data would be treated with a high level of confidentiality. Five parents responded positively. The ethical guidelines of the Norwegian Social Science Data Services, which gave permission for the project, have been satisfied.

**Informed Consent Statement:** Informed consent was obtained from all subjects involved in the study.

**Data Availability Statement:** The data presented in this study are not available due to participant privacy.

**Acknowledgments:** We would like to thank all the families involved in this project.

**Conflicts of Interest:** The authors declare no conflict of interest.

## Appendix A

The Cognitive Map is a conceptual framework that helps with analysing the interaction between a learner's cognitive acts and the elements of the activity that are relevant to these acts [51]. This process of analysing a task for identifying the cognitive operations is called "task analysis" [51,54]. As a part of the Workshop, a Cognitive Map associated with the activity "matching the figures" (Table A1) and one associated with "Matching and pairing socks" (Table A2) were provided to each parent.

**Table A1.** Cognitive Map associated with the activity "matching the figures".

| The Cognitive Map for "Matching Figures" | |
|---|---|
| Content | Figurative representations of animals. The images are within the daily experience of the child in ECEC institutions and in his/her reality (children encounter these types of animals in their everyday life in Norway). However, some details may be unfamiliar. |
| Modality | Figurative, using a motor response and with motor and verbal elements in elaboration and output. |
| Phases | Input:<br>CF1: Systemic exploration of a learning situation.<br>CF2: Precise and accurate understanding of words and meaning.<br>CF3: Well-developed understanding of the space concept.<br>CF5: Ability to use information from more than one source.<br>Elaboration:<br>CF6: Ability to experience and define a problem.<br>CF8: Ability to automatically make comparisons.<br>CF9: Ability to select relevant cues.<br>Output:<br>CF13: Controlled and planned expression of thought and actions.<br>CF14: Precise and accurate use of words and concepts. |
| Cognitive Operations | Comparison, identification, analysis, differentiation, analogical thinking, parent–child cooperation. |
| Level of novelty or complexity | Low to moderate. The complexity can rise by increasing the number of figures or by choosing unknown figures. |
| Level of Abstraction | Moderate to Low. The figures are simple and familiar. The level of abstraction can be increased by using more abstract figures or syllables/words. |
| Level of efficiency | The level of efficiency is assessed through two criteria: one objective (speed and precision in execution) and one subjective (personal effort employed in carrying out the task). In this activity the child is required to be accurate but not fast. The level of efficiency in this activity can be defined as moderate, but it will increase as the level of complexity or abstraction rises. |

**Table A2.** Cognitive Map associated with the activity "Matching and pairing socks".

| The Cognitive Map for "Matching and Pairing Socks" | |
|---|---|
| Content | Figurative representations of animals connected with real material (the socks). The images are within the daily experience of the child in ECEC institutions, and the activity is an activity that the child sees in the home environment during the tidying up and normal activities. However, some details may be unfamiliar. |
| Modality | Figurative, using a motor response and with motor and verbal elements in elaboration and output. Tactile in relation to the socks. |
| Phases | Input:<br>CF1: Systemic exploration of a learning situation.<br>CF2: Precise and accurate understanding of words and meaning.<br>CF3: Well-developed understanding of the space concept.<br>CF5: Ability to use information from more than one source.<br>Elaboration:<br>CF6: Ability to experience and define a problem.<br>CF8: Ability to automatically make comparisons.<br>CF9: Ability to select relevant cues.<br>CF10: Ability to relate objects and events to previous and anticipated situations.<br>Output:<br>CF13: Controlled and planned expression of thought and actions.<br>CF14: Precise and accurate use of words and concepts. |

**Table A2.** *Cont.*

| The Cognitive Map for "Matching and Pairing Socks" | |
| --- | --- |
| Cognitive Operations | Comparison, identification, analysis, differentiation, analogical thinking, and parent–child cooperation. |
| Level of novelty or complexity | Low to moderate. The complexity can rise by increasing the number of figures or by choosing unknown figures. |
| Level of Abstraction | Moderate to Low. The figures are simple and familiar. The activity is tactile. |
| Level of efficiency | The level of efficiency is assessed through two criteria: one objective (speed and precision in execution) and one subjective (personal effort employed in carrying out the task). In addition, the ability to recall a competence learned in another situation is observed. In this activity, the child is required to be accurate but not fast. The level of efficiency in this activity can be defined as moderate, but it will increase as the level of complexity or abstraction rises. |

**Appendix B**

During the workshop the parents received a questionnaire.
*Interview guide*

(1) How old are you? And your son/daughter?
(2) How long have you studied MLE? What have you studied (FIE basic, FIE standard, Bright Start, LPAD, . . . )?
(3) What do you think about the 12 criteria of mediation? Do you think that some of them are more important than others in a home environment?
(4) Which strategies in your opinion have you used the most during the two activities? Were they same during both activities?
(5) Did you know from before the meaning of the term "transfer"?
(6) Do you think that the linguistic development of your child has affected your communication during these activities?
(7) What do you think about the second activity chosen among "daily activities"?
(8) What is your opinion about being involved in a workshop to discuss both the observations and the Cognitive Maps with the researchers?
(9) Do you have any suggestions about the project structure or about activities that could be presented to children for enriching and supporting their transfer competence?

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
