# Peer review of "Mothers’ Perception about Mediated Learning Strategies Used in the Home Environment for Supporting the Transfer Ability in Children with Down Syndrome: An Exploratory Investigation"

_disabilities, doi:10.3390/disabilities2020019_

Round 1

Reviewer 1 Report

  1. I am surprised not to find in the introductory section of the paper any reference to the pioneering work of Rynders and Horrobin in the late Sixties in training cohorts of parents of young children with Down syndrome in Minneapolis and Chicago to optimize their parenthood particularly regarding health, language, and cognitive issues, as well as to the later experimental and clinical work of Mahoney and colleagues in Cleveland, Ohio, on parent’s responsive teaching in early intervention.
  2. Unfortunate also particularly from a theoretical point of view is the absence of reference to the classical study of Rondal (1976) demonstrating that the mothers of children with Down syndrome remarkably adapt their language to the linguistic levels of their children from the early ages on and complexify it gradually till adolescence in close relation with the linguistic  progresses of the children, exactly, and to the same extent, as the parents of typically developing children. This is basic as it guarantees that the intervention involving parents of children with Down syndrome rests on adequate linguistic grounds and that parental language to the children with Down syndrome can be relied upon to promote language development in the children just as this is the case with typically developing children (an idea that was not terribly popular worldwide in the early Seventies and before).
  3. Given the small number of subjects, this study qualifies at best for the status of exploratory investigation. The children involved are very young and the task proposed is a very simple one. These limitations should be clearly acknowledged in the paper starting with its title.
  4. Even more importantly, I wonder whether the paper brings about sufficiently new data and information to deserve publication in an international forum. The results suggest that parents are happy to play “matching the pictures” with their children with Down syndrome. Actually, many (most) parents naturally and spontaneously perform commenting, labelling, matching activities, and the like in free-play ineraction with their children with Down syndrome. There are dozens of papers published on this topic (and no acknowledgement of them in this paper). There is no need to spend time training parents to do just that. In my view, a more relevant and more timely question that would need to be dealt with in mediational research regards the promotion of more sophisticated aspects of langage and cognitive development. Would most parents of older children with Down syndrome be willing and happy to take responsibility for more complex teaching tasks and is it realistically feasible to train them successfully in this particular perspective?    

Authors responses

Answers to the comments from reviewer 1.

1) The pioneering work of Rynders and Horribin has been considered for the definition of the state of the art and of the theory. Therefore, they have been introduced in the introduction (§ 1) and in the theoretical framework (§ 3.1). In particular, in the theoretical framework, these studies have been considered, and the motivation for choosing the Feuerstein approach is furnished. Reference introduced: “Rynders JE, Horrobin JM. Down Syndrome, Birth to Adulthood: Giving Families an EDGE: ERIC; 1996.”. Also the studies developed by Mahoney have been taken into account, both in the introduction and in the theoretical framework. Reference introduced: “Mahoney G, Perales F, Wiggers B, Herman BB. Responsive teaching: early intervention for children with Down syndrome and other disabilities. Down Syndrome Research and Practice. 2006;11(1):18-28.”

2) The studies conducted by Rondal have been considered for both the introduction and the discussion. Different references have been evaluated for developing a more relevant discussion: “Rondal J-A, Docquier L. Maternal speech to children with Down Syndrome: An update. The Journal of Speech and Language Pathology–Applied Behavior Analysis. 2006;1(3):218.” “Rondal JA. Exceptional language development in Down syndrome: Implications for the cognition-language relationship: Cambridge University Press; 1995.” “Rondal JA. Maternal speech to normal and Down's syndrome children matched for mean length of utterance: University of Minnesota; 1976”.

3) The indication about the “exploratory investigation” has been introduced in the title and in the whole article.

4) This comment related to the fact that data were poor and not relevant was considered carefully. Consequently, the article has been greatly revised. However, it has also become impossible to highlight changes in the text. The data presented in the revision are related to the second part of the project and are not the same presented in the original version. The previous data were connected to observations conducted by parents working on an easy activity such as “matching the figures.” In the revised article, this activity is now just the starting point. The whole article analyses the answers given by mothers at the end of the whole project. The mothers (i.e. those trained in the Feuerstein approach) were involved in presenting two activities to children (with the aim of sustaining children’s transfer ability) in a workshop with the researchers and in answering the questionnaire that is the object of the revised article. Suggestions coming from the reviewer have been added at the end of the article. For example, we evaluate that the article can actually open possibilities for parents of older children with Down Syndrome.

Reviewer 2 Report

Thank-you for the opportunity to read about your work. Parental efficacy regarding being "teachers" of their children with disability is an important area of study. I found the observational tool that parents used very interesting; also parents' reflections on their own observations.

I think there is the potential for this work to be of interest and value to those researching and working in the area but I found the manuscript, overall, difficult to follow and confusing. You are bringing together many very abstract ideas, and I couldn't follow your description/arguments very easily.

I also have a query about the ethics of the research. Although you write that ethical guidelines have been respected, was ethical clearance obtained from your institution (or the organisation that families were recruited from?). 

Authors responses

Answers to the comments from reviewer 2.

The article has been simplified with a focus not on the observations of single cognitive functions but on mothers’ opinions about being involved in a project where the focus is on sustaining children’s transfer skills.

In addition to the observations and video to a questionnaire answered by mothers, the ethical issues have been revised and described more precisely.

Round 2

Reviewer 1 Report

I believe that the paper as modified by the authors is now acceprable for publication. I would suggest them to enlarge the scope of this type of research in future work;

Reviewer 2 Report

There has been extensive review of this manuscript. The rationale for the importance of research into mothers' perceptions of mediational strategies in the family environment is presented clearly. The methodology for the research is outlined and the findings presented logically with data to support. 

Thank you for the opportunity to review. All the best with your future work.